# Harmonizing Multi-Site Multi-Sequence Brain MRI via Semantic-Guided Conditional Diffusion

## Abstract

Training robust AI models for brain MRI analysis typically requires large datasets, prompting many studies to aggregate multi-site data. However, this introduces unwanted variations due to differences in scanners and/or acquisition protocols. These non-biological variations (known as site effects) can significantly compromise the performance and generalizability of downstream deep learning models. While image-level harmonization has emerged as a promising solution, existing methods frequently demand paired data (e.g., scans of the same subject at different sites) or costly encoder-decoder networks to disentangle anatomical content from pre-defined imaging style (*e.g.*, intensity and contrast), which struggle to comprehensively capture diverse image styles. Moreover, existing methods cannot adapt well across different MRI sequences, limiting their scalability. This paper proposes a semantic-guided conditional diffusion (SGCD) framework for unpaired 3D multi-sequence MRI harmonization. The SGCD first trains a conditional diffusion model (CDM) to align multi-site multi-sequence MRIs into a unified, sequence-specific domain, reducing global site-related variations. It then fine-tunes the CDM for target-specific harmonization using a style loss derived from BiomedCLIP trained on medical imaging data. By capturing differences in disentangled semantic image style between the harmonized and target MRIs, this loss enables effective harmonization that preserves anatomical structure and does not require paired training data. We evaluate SGCD on $4,163$ T1/T2-weighted MRIs from three multi-site datasets, with results suggesting its superiority over several state-of-the-art methods across voxel-level comparison, downstream classification, and brain tissue segmentation tasks.

## 1 Introduction

Recent advancements in machine learning (ML) and deep learning (DL) have led to powerful models for neuroimaging analysis, tackling tasks such as brain tissue segmentation, disease classification, and longitudinal studies from MRI scans. The statistical power and robustness of these models depend on access to large-scale training data, which often necessitates the aggregation of multi-site MRI data (An et al., 2022; Tofts & Collins, 2011; Schnack et al., 2010). However, this strategy introduces site effects—deeply embedded non-biological variations from differences in scanner hardware, imaging protocols, and software—that can confound ML/DL models and undermine their training and generalization (Gadewar et al., 2024; Parida et al., 2024).

To address site effects in multi-site studies, various harmonization techniques have been proposed, which can be broadly categorized as feature-level or image-level methods. Feature-level methods (Fortin et al., 2018; Pomponio et al., 2020) typically utilize statistical methods to correct site-specific variance in pre-extracted radiomic features. This approach is primarily limited by its reliance on feature quality and lacks generalizability across diverse downstream tasks (An et al., 2022; Cackowski et al., 2023). Harmonizing raw image data is increasingly recognized as an effective strategy to improve generalizability. Existing image-level harmonization methods that use generative models, such as generative adversarial networks (GANs) for cross-domain image translation (Liu et al., 2021; Chang et al., 2022; Modanwal et al., 2020), often suffer from training

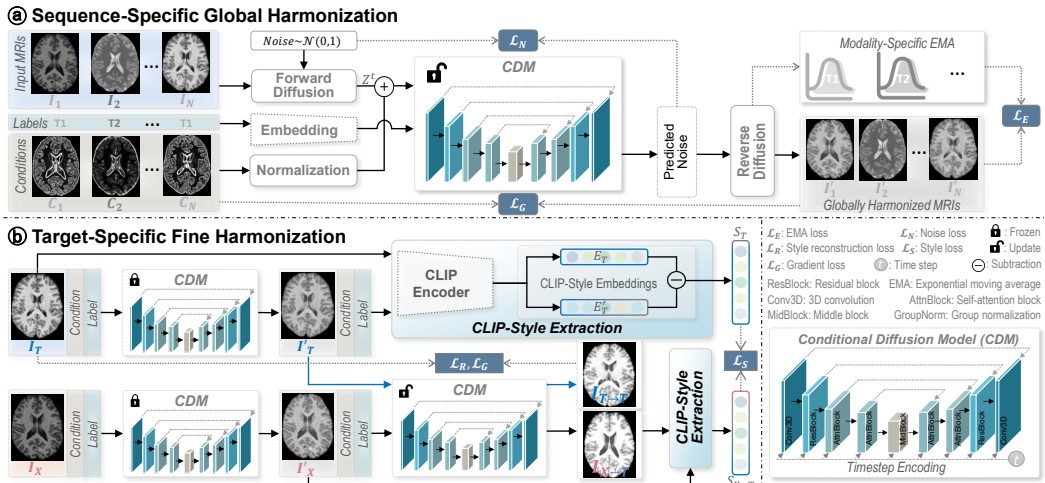

Figure 1: Overview of SGCD with two-stage training: (a) A conditional diffusion model (CDM) is first trained to globally harmonize multi-site multi-sequence MRIs into a sequence-specific unified domain, removing site-related global variations while preserving anatomical content; and (b) CDM is then fine-tuned to harmonize these globally aligned MRIs into the target style using a disentangled CLIP-style loss. This dual-stage process achieves efficient multi-site multi-sequence MRI harmonization without explicit content-style disentanglement.

instability and mode collapse. Some methods using variational autoencoders (VAEs) or multiple encoder-decoder networks aim to learn disentangled representations of anatomical and style information for harmonization (Zuo et al., 2021; Dewey et al., 2020; Cackowski et al., 2023), but they are typically computationally intensive. Another limitation of existing methods is their reliance on paired training data, such as scans from traveling subjects or multiple sequences (*e.g.*, T1- and T2-weighted MRIs) of the same subject. This requirement is often challenging to meet in large-scale retrospective studies, posing a practical barrier to their application.

To this end, we propose a new semantic-guided conditional diffusion framework (SGCD) for multi-sequence 3D brain MRI harmonization. As illustrated in Fig. 1, our SGCD adopts a two-stage progressive training scheme. The first *global harmonization* stage trains a site-agnostic conditional diffusion model (CDM) to align multi-sequence MRIs from all sites to a sequence-specific global domain, mitigating global differences (*e.g.*, intensity and contrast). This is achieved through style-free gradient map conditioning and losses based on exponential moving average (EMA) record, supporting multi-sequence input without requiring paired samples. The second *target-specific harmonization* stage fine-tunes this CDM to harmonize the globally aligned MRIs into a pre-defined target space using a semantic style loss. A pre-trained BiomedCLIP encoder (Zhang et al., 2024) is used to extract image styles by capturing the difference in semantic embeddings between the original target MRIs and their globally aligned counterparts. As these pairs share the same anatomical information, the difference in their semantic embeddings represents the target style, effectively disentangled from the content information. This dual-stage training scheme leverages the conditional generative power of diffusion models and the semantic-rich embeddings from BiomedCLIP, achieving content-style disentanglement during multi-sequence harmonization without requiring auxiliary encoder-decoder networks or paired training samples. The SGCD is trained and evaluated on three multi-site datasets with a total of 4,163 T1-weighted (T1w) or T2-weighted (T2w) MRIs through three tasks. Experimental results demonstrate the superiority of SGCD over several state-of-the-art (SOTA) methods in aligning multi-site and multi-sequence MRI styles while preserving critical biological and anatomical features.

## 2 RELATED WORK

Learning-based neuroimaging studies require large-scale datasets to enable robust training and statistical power (Dufumier et al., 2022; Zhu et al., 2023; Hawco et al., 2022). This is often achieved

by retrospectively pooling data across studies, which typically lack standardized scanning protocols. MRI data, however, are highly sensitive to site-related factors such as field strength, sequence design, and reconstruction software (Parida et al., 2024). These differences yield variations in intensity, contrast, and signal-to-noise ratio that, while negligible to radiologists, confound model training by entangling biological and site-specific features, resulting in unstable training and poor cross-site generalization.

Existing brain MRI harmonization methods fall into *feature-level* and *image-level* approaches. Feature-level methods, such as ComBat (Fortin et al., 2018), model site effects as additive and multiplicative biases within an empirical Bayes framework. While effective for specific radiomic features (e.g., gray matter volumes, cortical thickness), they are constrained by feature quality and lack flexibility for diverse downstream tasks (An et al., 2022; Cackowski et al., 2023). Image-level methods operate directly on raw MRIs, offering broader applicability. Basic techniques such as min-max, z-score, WhiteStripe, and histogram matching standardize intensity distributions but only capture global differences, leaving spatially varying contrast and noise uncorrected.

Recent ML/DL-based approaches treat image-level harmonization as a generative task akin to image translation or style transfer. Early methods relied on paired data for supervision, using *traveling subjects* scanned across sites (Xu et al., 2024) or paired multi-sequence MRIs (e.g., T1, T2, FLAIR) to disentangle anatomy from contrast (Zuo et al., 2021; Dewey et al., 2020; Zuo et al., 2023). However, such paired data are costly and impractical for large-scale retrospective studies.

Recent research emphasizes unpaired harmonization, often using CycleGAN (Modanwal et al., 2020; Liu et al., 2021) or normalizing flows (Beizaee et al., 2023) to map scanner domains. Other methods disentangle anatomy and style, either by unlearning site/scanner effects (Cackowski et al., 2023) or by exploiting weak supervision from multi-view 2D slices (Zuo et al., 2022). However, these approaches face key limitations: (1) disentanglement typically requires multiple encoder–decoder networks and latent code swapping, which is computationally costly (Ouyang et al., 2021); (2) reliance on 2D slices neglects 3D spatial context, causing artifacts; and (3) Many are sequence-specific and require retraining for new sequences or sites, reducing practicality. To address these issues, we propose a conditional diffusion model with semantic-rich MRI embeddings from a pre-trained BiomedCLIP encoder, enabling effective multi-sequence harmonization without paired data or auxiliary networks.

# 3 METHODOLOGY

### 3.0.1 PROBLEM FORMULATION.

The goal of image-level harmonization is typically to either transform multi-site MRIs into a unified virtual domain (Xu et al., 2024) or harmonize all source MRIs into a pre-selected target domain derived from one or more reference scans (Cackowski et al., 2023; Zuo et al., 2022; Wu et al., 2025). Our SGCD framework adopts a progressive two-stage training scheme capable of achieving both objectives. In the first global harmonization stage, all input MRIs, across multiple sequences and sites, are harmonized into a sequence-specific unified domain, eliminating global site-related intensity variations. In the second target-specific harmonization stage, all globally aligned MRIs are further transformed into a pre-selected target domain.

## 3.1 STAGE 1: SEQUENCE-SPECIFIC GLOBAL HARMONIZATION

To reduce site effect from multi-site and multi-sequence acquisitions, the first stage aims to normalize global intensity variations while preserving anatomical structure. Unlike traditional site-specific training, we propose a *sequence-specific global harmonization approach* by training a conditional diffusion model (CDM) across all sites and sequences simultaneously. The CDM is trained by two key mechanisms: (1) the sequence-specific, style-free conditions, obtained from sequence labels and normalized gradient maps; and (2) the exponential moving average (EMA)-based record update, which serves as a dynamic harmonization target, ensuring the model aligns MRIs of each sequence to a stable unified domain. This formulation enables better generalization to unseen domains, eliminates the need for site-specific models, and establishes a unified intermediate representation that aids the subsequent target-specific harmonization.

### 3.1.1  1) CONDITIONAL DIFFUSION MODEL:

As shown in Fig. 1 (a), this stage trains a conditional diffusion model $\Phi$ that takes N multi-site MRIs $\{I_i\}_{i=1}^N \in \mathbb{R}^{1 \times W \times H \times D}$ as input. Each MRI $I_i$ is paired with a sequence label $m_i \in \{1, \cdots, M\}$ drawn from a set of $M$ classes to differentiate the MRI sequence type (*e.g.*, 1 for T1w, 2 for T2w). Here $W$, $H$, and $D$ denote the width, height, and depth of the 3D MRI volume, respectively. The input then goes through a forward diffusion process (FDP) governed by a Markov chain with a total of $T$ timesteps. During FDP, noise is sampled from a standard Gaussian distribution and gradually added to $I_i$ to create a noisy image $I_i^t$ at each timestep $t$:

$$I_i^t = \sqrt{\bar{\alpha}_t} I_i^0 + \sqrt{1 - \bar{\alpha}_t}\epsilon, \quad \epsilon \sim \mathcal{N}(\mathbf{0}, \boldsymbol{I}), \tag{1}$$

where $\epsilon$ is the sampled noise, $\bar{\alpha}_t := \prod_{i=1}^t \alpha_i$, $\alpha_t := 1 - \beta_t$, and $\beta_t$ follows a predefined variance schedule (Ho et al., 2020). To preserve anatomical structure of the brain, we use each MRI's gradient map $G_{i=1:N} \in \mathbb{R}^{1 \times W \times H \times D}$ as the input condition to the CDM, computed as:

$$G(I_i) = Pad(\frac{1}{3}(\nabla_H I_i + \nabla_W I_i + \nabla_D I_i)), \tag{2}$$

where $\nabla$ is the forward-difference operator along each spatial axis, and $Pad(\cdot)$ denotes zero padding to restore the original input size. Each gradient map is normalized to $[-1, 1]$. This style-free anatomical condition is concatenated with the noisy image $I_i^t$ and fed into CDM, while the embedded sequence label $m_i$ serves as the class condition. CDM is implemented as a time-conditioned 3D U-Net and trained to predict the noise $\epsilon_\theta$ by minimizing the noise-level loss:

$$\mathcal{L}_N = \|\epsilon - \epsilon_\theta(I_i^t, t, G_i, m_i)\|_2^2. \tag{3}$$

During training, we also derive $\hat{I}_i'$, which is an intermediate, one-step estimate of the final globally harmonized MRIs $I_i'$ through one reverse diffusion process (RDP):

$$\hat{I}_i' \approx I_i' = \frac{1}{\sqrt{\bar{\alpha}_t}}(I_i^t - \sqrt{1 - \bar{\alpha}_t}\epsilon_\theta(I_i^t, t, G_i, m_i)), \tag{4}$$

### 3.1.2  2) EMA-BASED RECORD UPDATE:

We then use $\hat{I}_i'$ to update a sequence-specific exponential moving average (EMA) record to guide the global alignment across different sites. The EMA record for each sequence contains a fully differentiable soft histogram, mean, and standard deviation (std.) of intensity values. Let $x \in \mathbb{R}^F$ be a 1D flattened tensor of $\hat{I}_i'$; we compute a soft-histogram over a fixed value range $[v_{min}, v_{max}]$ with $K$ bins. The bin centers can be defined as $c_k = v_{\min} + \frac{k-1}{K-1}(v_{\max} - v_{\min})$ for $k = 1 : K$. For each voxel value $x_i$, its contribution to the $k$-th bin is computed using a Gaussian kernel:

$$w_{ik} = \exp\left(-\frac{1}{2\sigma^2}(x_i - c_k)^2\right), \tag{5}$$

where $\sigma$ is the kernel bandwidth controlling smoothness. The normalized soft-histogram $\mathcal{H}(x) \in \mathbb{R}^K$ is defined by its $k$-th components as:

$$\mathcal{H}_k(x) = \frac{\sum_{i=1}^F w_{ik}}{\sum_{j=1}^K (\sum_{i=1}^F w_{ij}) + \delta}, \quad \text{for } k = 1, \cdots, K, \tag{6}$$

where $\delta$ is a small constant to avoid division by zero. We denote the differentiable soft-histogram of $\hat{I}_i'$ as $\mathcal{H}(\hat{I}_i')$. We compute the soft-histogram for each $\hat{I}_i'$ and update the corresponding EMA record for the $m$-th sequence as follows:

$$EMA_m^{\{\mathcal{H}\}} = \gamma \cdot EMA_m^{\{\mathcal{H}\}} + (1 - \gamma)\mathcal{H}(\hat{I}_i'), \tag{7}$$

where $\gamma \in [0, 1)$ is the EMA decay factor controlling the update rate. We update the $EMA_m^{\{\mu\}}$ and $EMA_m^{\{\sigma\}}$ for the voxel mean and std. intensities similarly. This EMA record serves as the running estimate of the sequence-specific intensity statistics. After every EMA update, we calculate an EMA loss to guide the global harmonization, defined as:

$$\mathcal{L}_E = WD(EMA_m^{\{\mathcal{H}\}}, \mathcal{H}(\hat{I}_i')) + \|EMA_m^{\{\mu\}} - \mu(\hat{I}_i')\|_2^2 + \|EMA_m^{\{\sigma\}} - \sigma(\hat{I}_i')\|_2^2, \tag{8}$$

where the $WD(\cdot)$ is the differentiable Wasserstein distance, which is computed as the mean absolute difference between the cumulative distribution functions of two soft-histograms.

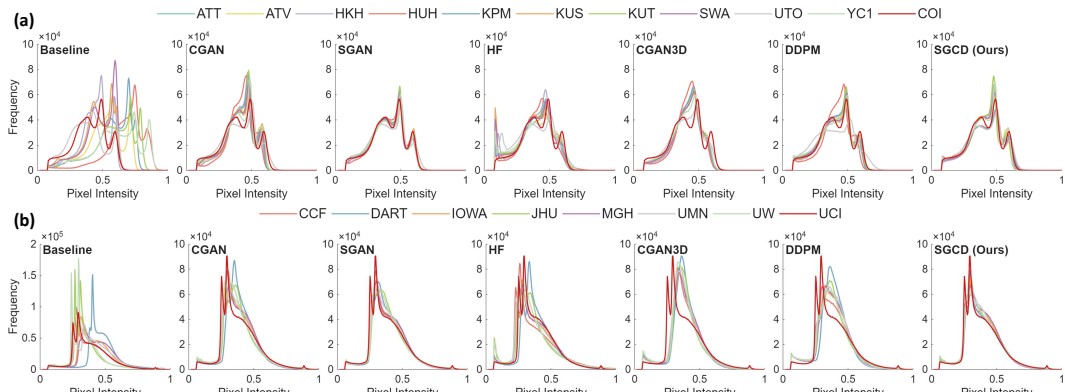

Figure 2: Histograms of (a) 22 T1w MRIs from SRPBS test set across 11 sites, with COI as the target domain; (b) 16 T2w MRIs from DWI-THP test set across 8 sites, with UCI as the target domain.

### 3.1.3 ANATOMY PRESERVATION CONSTRAINT:

To preserve brain anatomical structures during harmonization, we compute the normalized gradient map $G_i$ of $I'_i$, and define a gradient loss conditioned on $G_i$ as: $\mathcal{L}_G = \|G_i - G(\hat{I}'_i)\|_2^2$. The hybrid loss for Stage 1 is defined as: $\mathcal{L}_1 = \mathcal{L}_N + \mathcal{L}_G + \mathcal{L}_E$. This first stage training creates a global harmonizer capable of aligning multi-site, multi-sequence MRIs simultaneously to unified domains, achieving target-free harmonization.

## 3.2 STAGE 2: TARGET-SPECIFIC FINE HARMONIZATION

To adapt globally harmonized MRIs to a specific target style, the second stage fine-tunes the pre-trained CDM from Stage 1 using unpaired source and target data of the same sequence. This stage builds upon the site-agnostic intermediate representations generated earlier, introducing a novel style guidance derived from a pre-trained vision-language model BiomedCLIP (Zhang et al., 2024). The fine-tuning is guided by two key principles: (1) a diffusion-based translation process, which iteratively transforms globally harmonized MRIs to the target style; and (2) a disentangled CLIP-style loss, which measures style differences in a semantic embedding space, removing the need for paired data or explicit style-content separation. This formulation not only achieves efficient target-specific harmonization while preserving anatomy and sequence specificity but also offers improved generalizability on unseen domains.

As shown in Fig. 1 (b), given multi-sequence source MRIs $I_X$ and unpaired target MRIs $I_T$ of the same sequence, we first globally harmonize them into $I'_X$ and $I'_T$ using the pre-trained CDM from the first stage. We then fine-tune the CDM to translate $I'_X$ into a target-style MRI $I_{X \to T}$ guided by a pre-trained BiomedCLIP encoder (Zhang et al., 2024) that implicitly extracts style embeddings. We introduce a *hybrid disentangled CLIP-style loss* to achieve this translation without requiring paired data or explicit style definitions. The globally harmonized $I'_X$ and $I'_T$ are obtained through CDM inference that employs the DDIM sampling strategy (Song et al., 2020) on RDP. Instead of the one-step estimate $\hat{I}'_i$ used in training (see Eq. 4), we obtain $I'_i$ by iteratively denoising over $T_r$ steps ($t = T_r : 0$):

$$I_i^{t-1} = \sqrt{\bar{\alpha}_{t-1}}\hat{I}'_i + \sqrt{1 - \bar{\alpha}_{t-1}}\epsilon_\theta(I_i^t, t, G_i, m_i), \tag{9}$$

where $\hat{I}'_i$ is the intermediate estimate used during training and $\epsilon_\theta$ is the trained CDM function.

### 3.2.1 TARGET-SPECIFIC FINE-TUNING:

The CDM is then fine-tuned to adapt $I'_X$ to match the style of $I_T$ through the DDIM sampling strategy, containing an FDP followed by an RDP. Unlike Eq. 1, where random noise is added to $I_i$, we now iteratively add CDM-generated noise to $I'_X$ over $T_f$ forward iterations ($t = 0 : T_f$) to get the noisy image $I_X^{T_f}$:

$$I_X^{t+1} = \sqrt{\bar{\alpha}_{t+1}}I'_X + \sqrt{1 - \bar{\alpha}_{t+1}}\epsilon_\theta(I_X^t, t, G_X, m_X), \tag{10}$$

We then apply RDP to denoise $I_X^{T_f}$ back to the final translated MRI over $t = T_r : 0$ steps:

$$I_X^{t-1} = \sqrt{\bar{\alpha}_{t-1}} I_X' + \sqrt{1 - \bar{\alpha}_{t-1}} \epsilon_\theta(I_X^t, t, G_X, m_X), \tag{11}$$

After $T_r$ timesteps, we obtain the harmonized MRI: $I_{X \to T}$.

### 3.2.2 DISENTANGLED CLIP-STYLE GUIDANCE:

To guide style translation without requiring paired data or explicit style learning, we incorporate a pre-trained BiomedCLIP encoder to extract implicit style representations. Specifically, we compute style embeddings as the difference in CLIP-space between each MRI and its globally-aligned counterpart:

$$S_T = \Psi(I_T) - \Psi(I_T'), \quad S_{X \to T} = \Psi(I_{X \to T}) - \Psi(I_X'), \tag{12}$$

where $\Psi(\cdot)$ is the BiomedCLIP image encoder. Since $I_T'$ shares content with $I_T$, the difference $S_T$ captures target-specific style, disentangled from content information. Similarly, $S_{X \to T}$ reflects the style of the harmonized source. We then define the *style translation loss* as:

$$\mathcal{L}_S = \|S_T - S_{X \to T}\|_1 + (1 - \frac{S_T \cdot S_{X \to T}}{\|S_T\|\|S_{X \to T}\|}), \tag{13}$$

where the 1st term is the $l_1$ distance in the CLIP-embedding space and the 2nd term quantifies the directional discrepancy between two style embeddings.

To ensure style consistency, we further design a *style reconstruction loss* by minimizing style embeddings of each target MRI and its harmonized counterpart: $\mathcal{L}_R = \|S_T - S_{T \to T}\|_1$. The hybrid disentangled CLIP-style loss is defined as: $\mathcal{L}_2 = \mathcal{L}_S + \mathcal{L}_R$.

By leveraging BiomedCLIP's semantic-rich embeddings, the CDM effectively translates source MRIs to the target style without requiring paired training data or explicit image style and content disentanglement learning, ensuring that anatomical content remains unchanged.

### 3.2.3 ADAPTATION TO NEW DATA.

The proposed two-stage training strategy improves the generalizable potential of SGCD. When MRIs from an unseen site are used as the source domain, they can be harmonized without retraining the model. As the global harmonizer, trained across multiple sites and sequences, generalizes to new domains by aligning input MRIs to a learned site-agnostic intermediate representation. The target-specific harmonizer then maps these globally harmonized MRIs into a learned target style. If an unseen site is used as the target domain, only the second-stage model requires fine-tuning to learn the new domain-specific style, while the first-stage harmonizer remains unchanged.

### 3.2.4 IMPLEMENTATION.

We use the MONAI (Cardoso et al., 2022) framework to implement the proposed SGCD approach. CDM is implemented as a time-conditioned 3D U-Net with a symmetric architecture comprising 2 upsampling/downsampling layers, 2 residual blocks, 4 self-attention blocks, and a middle block, with channels $\{32, 64, 256, 256\}$, respectively. Stage 1 training uses the default Adam optimizer with an initial learning rate (LR) of $1 \times 10^{-4}$, while Stage 2 fine-tuning uses an LR of $1 \times 10^{-6}$. For EMA-based histogram/record updates, we set the decay factor $\gamma = 0.2$ and compute soft histograms with $K = 100$ bins over the range $[v_{\min} = 0, v_{\max} = 1]$. The variance scheduler $\beta$ is empirically set to increase linearly from 0.0015 to 0.0195. We apply $T = 1,000$ noise steps for the global harmonization stage and empirically set $T_f = 35$ and $T_r = 25$ for the target-specific harmonization stage via grid search. See Appendix A for detailed implementations.

## 4 EXPERIMENT

### 4.0.1 STUDIED MATERIALS.

Three brain MRI datasets are utilized: (1) OpenBHB (Dufumier et al., 2022), with T1w MRIs from 3,984 healthy subjects across 58 sites; (2) SRPBS (Tanaka et al., 2021), with T1w MRIs from 9 traveling subjects across 11 sites; and (3) DWI-THP (Magnotta et al., 2012) with T1w and T2w

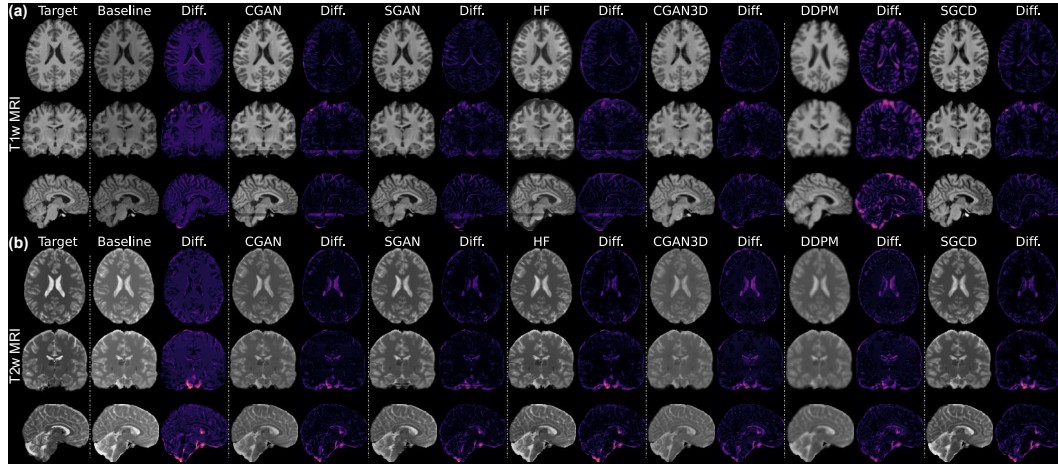

Figure 3: Visualization of harmonization results for (a) a T1-weighted MRI sample from SRPBS and (b) a T2-weighted sample from DWI-THP. Each panel shows the original source MRI (Baseline), harmonized source MRIs from six different methods, and the corresponding difference (Diff.) maps between the harmonized source and target MRIs.

Table 1: Comparison between source site MRIs and corresponding target site (COI) MRIs with matching subjects on the SRPBS test set (2 subjects across 11 sites).

| Method | SSIM ↑ | PSNR ↑ | PCC ↑ | WD ↓ |
|---|---|---|---|---|
| Baseline | $0.881^{0.03}$ | $22.03^{4.29}$ | $0.980^{0.01}$ | $0.041^{0.03}$ |
| CGAN | $0.888^{0.03}$ | $27.64^{1.64}$ | $0.976^{0.01}$ | $0.006^{0.002}$ |
| SGAN | $0.894^{0.02}$ | $27.33^{1.51}$ | $0.978^{0.01}$ | $\mathbf{0.005}^{0.002}$ |
| HF | $0.850^{0.02}$ | $26.38^{1.55}$ | $0.969^{0.01}$ | $0.008^{0.002}$ |
| CGAN3D | $0.866^{0.02}$ | $27.55^{1.48}$ | $0.976^{0.01}$ | $0.009^{0.003}$ |
| DDPM | $0.800^{0.10}$ | $25.39^{3.50}$ | $0.950^{0.03}$ | $0.006^{0.001}$ |
| SGCD (Ours) | $\mathbf{0.896}^{0.02}$ | $\mathbf{28.04}^{1.71}$ | $\mathbf{0.988}^{0.01}$ | $\mathbf{0.005}^{0.001}$ |

MRIs from 5 subjects scanned across 8 sites. The OpenBHB is split into a training set (3,227 MRIs) and a validation set (757 MRIs). For SRPBS, we use 66 MRIs from 6 subjects across 11 sites for training, 11 MRIs from 1 subject for validation, and 22 MRIs from 2 subjects for testing. For DWI-THP, 48 MRIs from 3 subjects across 8 sites (both modalities) are used for training, and the remaining 32 MRIs for testing. Given the lack of standardized criteria for target site selection in brain MRI harmonization, we follow (Tian et al., 2022) and select the site exhibiting the *lowest intra-site style variability* for each dataset, measured by the mean Wasserstein Distance (WD) across samples.

### 4.0.2 COMPETING METHODS.

We compare our SGCD with five SOTA image-level MRI harmonization methods: Cycle-GAN (**CGAN**) (Chang et al., 2022), StyleGAN (**SGAN**) (Liu et al., 2021), 3D CycleGAN (**CGAN3D**) (Zhu et al., 2017), Harmonizing Flow (**HF**) (Beizaee et al., 2023), and **DDPM** (Durrer et al., 2023). The 3D methods (*i.e.*, CGAN3D and DDPM) are trained using the same volumetric dataset as our method, while the 2D methods (*i.e.*, CGAN, SGAN, and HF) use axial MRI slices derived from the same dataset. We ensured consistent training hyperparameters across all methods for fair comparisons.

### 4.0.3 TASK 1: HISTOGRAM & VOXEL-LEVEL COMPARISON.

This experiment utilizes SRPBS and DWI-THP with traveling subjects. For SRPBS, each method harmonizes MRIs from 10 source sites to a target site (ID: COI). For DWI-THP, harmonization is performed from 7 source sites to a target site (ID: UCI). Performance is assessed via histogram comparisons and voxel-level metrics: PSNR, structural similarity index (SSIM), Wasserstein distance (WD), and Pearson correlation coefficient (PCC) with raw source MRIs as **Baseline**. Our method is trained end-to-end across all T1w and T2w MRIs in DWI-THP simultaneously, while

Figure 4: Segmentation maps with White: accurate segmentation; Red: under-segmentation; Blue: over-segmentation.

Table 2: Results (%) achieved by different methods in terms of AP and DSC metrics for gray matter (GM) and white matter (WM) segmentation on DWI-THP.

| Method | AP ↑ | | | DSC ↑ | | |
|--------|------|------|------|-------|------|------|
| | GM | WM | Mean | GM | WM | Mean |
| CGAN | $96.1^{1.3}$ | $96.0^{0.6}$ | $96.1^{0.9}$ | $88.0^{0.9}$ | $92.0^{0.3}$ | $90.0^{0.6}$ |
| SGAN | $95.7^{0.7}$ | $97.1^{0.6}$ | $96.4^{0.4}$ | $87.9^{0.8}$ | $92.0^{0.3}$ | $90.0^{0.5}$ |
| HF | $94.5^{1.3}$ | $95.2^{0.8}$ | $94.8^{0.9}$ | $87.8^{1.0}$ | $91.7^{0.4}$ | $89.7^{0.6}$ |
| CGAN3D | $86.2^{1.5}$ | $96.0^{0.6}$ | $91.1^{0.7}$ | $82.8^{0.9}$ | $90.1^{0.2}$ | $86.4^{0.5}$ |
| DDPM | $92.9^{1.4}$ | $91.9^{1.4}$ | $92.4^{1.3}$ | $74.4^{8.3}$ | $79.0^{9.8}$ | $76.7^{9.0}$ |
| SGCD (Ours) | $\mathbf{97.8}^{0.4}$ | $\mathbf{99.6}^{0.3}$ | $\mathbf{98.7}^{0.2}$ | $\mathbf{93.0}^{0.7}$ | $\mathbf{94.7}^{0.2}$ | $\mathbf{93.8}^{0.4}$ |

the competing deep models require separate training for each sequence. As shown in Fig. 2, both datasets exhibit substantial site-wise intensity variations (Baseline). Without relying on explicit style learning or retraining for each sequence, our method effectively aligns source MRI histograms to those of the target site across both sequences. Additional results are presented in Appendix A.

As shown in Table 1, SGCD achieves the highest SSIM, PSNR, and PCC, indicating superior voxel-level agreement and anatomy preservation. It also achieves the best WD score (tied with SGAN), confirming effective style alignment. The visualization results in Fig. 3 show that SGCD-harmonized MRIs more closely resemble the target domain for T1w and T2w sequences, while the 2D methods (CGAN, SGAN, and HF) generate strip artifacts in different views. These results highlight SGCD's ability to harmonize MRIs while maintaining high image quality and anatomy fidelity.

### 4.0.4 TASK 2: BRAIN TISSUE SEGMENTATION.

We further evaluate anatomy preservation via a brain tissue segmentation task on T1 MRIs in DWI-THP. FreeSurfer (Billot et al., 2023) is used to generate gray matter (GM) and white matter (WM) segmentation maps for original and harmonized MRIs. Segmentation quality is assessed using the Anatomical Preservation (AP) score (Parida et al., 2024), which measures the relative absolute difference in tissue volumes, and Dice Similarity Coefficient (DSC), which quantifies the spatial overlap between segmentation maps. Table 2 shows that SGCD achieves the highest AP and DSC for both GM and WM, and mean scores. Figure 4 further shows that SGCD yields fewer segmentation errors in the WM and GM tissue boundaries. This superior anatomical fidelity may be attributed to our gradient-based anatomy conditioning and constraint, and the implicit content-style disentanglement in CLIP-style guidance during the target-specific fine-tuning stage.

Table 3: Site classification and age prediction results on original (Baseline) and harmonized MRIs on OpenBHB.

| Method | Site Classification (%) | | | | Age Prediction | |
|--------|------|------|------|------|------|------|
| | BAC ↓ | F1 ↓ | PRE ↓ | Recall ↓ | MAE ↓ | MSE ↓ |
| Baseline | $34.3^{2.40}$ | $66.3^{2.30}$ | $75.7^{1.80}$ | $73.2^{1.90}$ | $5.30^{0.260}$ | $47.4^{1.41}$ |
| CGAN | $42.5^{1.60}$ | $69.5^{2.70}$ | $77.0^{3.00}$ | $73.9^{2.00}$ | $6.63^{0.264}$ | $79.0^{10.5}$ |
| SGAN | $25.8^{2.20}$ | $59.3^{1.20}$ | $66.2^{1.50}$ | $65.1^{1.50}$ | $7.31^{0.494}$ | $85.7^{12.9}$ |
| HF | $34.2^{1.10}$ | $66.5^{2.00}$ | $73.6^{2.10}$ | $72.3^{2.10}$ | $5.84^{0.221}$ | $57.0^{3.85}$ |
| CGAN3D | $32.4^{2.90}$ | $65.6^{1.90}$ | $75.1^{1.90}$ | $72.3^{1.70}$ | $5.90^{0.360}$ | $\mathbf{33.3}^{9.67}$ |
| DDPM | $16.6^{1.60}$ | $56.0^{1.30}$ | $\mathbf{54.5}^{0.70}$ | $53.5^{2.00}$ | $5.33^{0.258}$ | $48.5^{6.80}$ |
| SGCD | $\mathbf{14.5}^{1.30}$ | $\mathbf{54.3}^{2.50}$ | $56.0^{1.80}$ | $\mathbf{52.5}^{3.14}$ | $\mathbf{5.24}^{0.141}$ | $54.0^{2.02}$ |

### 4.0.5 TASK 3: SITE CLASSIFICATION & BRAIN AGE PREDICTION.

We evaluate SGCD in reducing site-related variations through two downstream tasks (*i.e.*, site classification and brain age prediction). Each method is trained on OpenBHB training data and applied

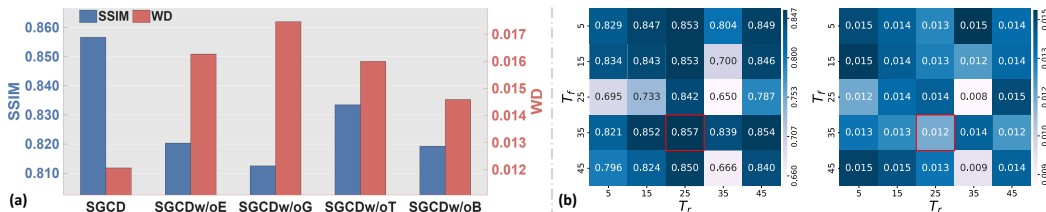

Figure 5: Results of (a) SGCD with 4 variants on DWI-THP and (b) with different parameters.

to harmonize the validation data with Site 17 as the target site. We extract deep features from these harmonized MRIs using ResNet18 (He et al., 2016) (with its final layer removed). A logistic regression, trained on 70% of these deep features and tested on 30%, performs multiclass site classification, and a ridge regressor predicts brain age. Both tasks are repeated 5 times for random data partition, with mean and standard deviation results reported in Table 3. Lower site classification results indicate better removal of site-related variations, while lower age prediction error suggests superior anatomical feature preservation. Table 3 shows that SGCD yields the lowest site classification performance in BAC, F1, and Recall, effectively removing site-related features and maintaining faithful anatomical integrity with lower mean absolute error (MAE).

## 5 DISCUSSION

### 5.0.1 ABLATION STUDY.

We assess four key components of SGCD by comparing it to its four variants: **SGCDw/oE** (without EMA-based constraint), **SGCDw/oG** (without gradient-based anatomical condition and constraint), **SGCDw/oT** (without Stage 2), and **SGCDw/oB** that uses style embeddings from CLIP pretrained on natural images (Radford et al., 2021) (rather than from BiomedCLIP pretrained on medical data). As shown in Fig. 5 (a), SGCDw/oE shows degraded performance (lower SSIM, higher WD), indicating the EMA constraint is critical for globally aligning multi-sequence inputs into their unified domains. SGCDw/oG performs the worst across all metrics, highlighting the necessity of gradient-based anatomical conditioning for maintaining structural fidelity and facilitating content-style disentanglement. SGCDw/oT aligns global intensity but fails to capture local style features such as tissue contrast and texture, confirming the importance of the target-specific stage. SGCDw/oB shows moderate WD but compromises anatomical integrity. This implies that CLIP trained on natural images lacks domain-specific knowledge to capture subtle variations in medical imaging, such as site-related style differences. In contrast, SGCD employs BiomedCLIP, pretrained on the PMC-15M dataset of medical image–text pairs with diverse modalities, like MRI, CT, and X-ray (Zhang et al., 2024), enabling domain-aware harmonization that preserves anatomy while adapting site- and sequence-specific styles.

### 5.0.2 INFLUENCE OF DIFFUSION PARAMETERS.

We perform a grid search over the forward ($T_f$) and reverse ($T_r$) diffusion steps, evaluating volume-level metrics on the DWI-THP test set. As shown in Fig. 5 (b), the model performs best when $T_f$ and $T_r$ are comparable. While some combinations yield lower WD scores, SGCD with $T_f = 35$ and $T_r = 25$ offers good image quality by prioritizing anatomical fidelity.

## 6 CONCLUSION

We present a semantic-guided conditional diffusion (SGCD) framework for multi-site multi-sequence MRI harmonization. SGCD first aligns MRIs into a unified, sequence-specific space via style-free gradient conditioning, then performs target-specific harmonization using CLIP-based semantic style embeddings, enabling effective volume-level harmonization without paired data or explicit style learning. Evaluated on three multi-site datasets, SGCD outperforms SOTA methods in removing site-related variations while preserving anatomical fidelity across T1w and T2w MRIs. Future work will extend SGCD to MRIs with pathological features such as lesions and tumors.

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

# A APPENDIX

# Harmonizing Multi-Site Multi-Sequence Brain MRI via Semantic-Guided Conditional Diffusion
# – *Supplementary Materials*

**Anonymous authors**

## 1 SGCD Algorithm Implementation

Here we include a brief description of the algorithm of the first sequence-specific global harmonization stage and the second target-specific fine harmonization stage. The global harmonization is trained for $N_1 = 300$ epochs and the target-specific harmonization is fine-tuned for $N_2 = 30$ epochs.

---

**Algorithm 1** Sequence-Specific Global Harmonization

---

**Input**: Multi-site MRIs with sequence label $\{I_i, m_i\}_{i=1}^N$
**Parameter**: Number of epochs $n_{\text{epochs}}$, EMA decay $\gamma$, histogram bins $K$, value range $[v_{\min}, v_{\max}]$, diffusion steps $T$
**Output**: Trained conditional diffusion model (CDM) $\Phi$

1: Initialize $\Phi$ (3D U-Net), optimizer, and EMA records for each sequence $m$: $\text{EMA}_m^{\{\mathcal{H}\}}, \text{EMA}_m^{\{\mu\}}, \text{EMA}_m^{\{\sigma\}}$
2: **for** epoch $= 1$ to $N_1$ **do**
3:     Compute normalized gradient map $G_i = \text{Norm}(\nabla(I_i))$ for each $I_i$
4:     Sample random timestep $t \sim \{1, \ldots, T\}$
5:     Sample noise $\epsilon \sim \mathcal{N}(0, I)$
6:     Generate noisy image $I_i^t = \sqrt{\bar{\alpha}_t} I_i + \sqrt{1 - \bar{\alpha}_t} \epsilon$
7:     Concatenate $I_i^t$ and $G_i$ as input, embed $m_i$ as class condition
8:     Predict noise: $\hat{\epsilon} = \Phi(I_i^t, t, G_i, m_i)$
9:     **Noise loss:** $\mathcal{L}_N = \|\epsilon - \hat{\epsilon}\|_2^2$
10:     Estimate harmonized MRI: $I_i' = \frac{1}{\sqrt{\bar{\alpha}_t}}(I_i^t - \sqrt{1 - \bar{\alpha}_t} \hat{\epsilon})$
11:     Compute soft-histogram $\mathcal{H}(I_i')$, mean $\mu(I_i')$, std $\sigma(I_i')$ over $I_i'$
12:     **EMA update for sequence** $m_i$**:**
13:         $\text{EMA}_{m_i}^{\{\mathcal{H}\}} \leftarrow \gamma \cdot \text{EMA}_{m_i}^{\{\mathcal{H}\}} + (1 - \gamma)\mathcal{H}(I_i')$
14:         $\text{EMA}_{m_i}^{\{\mu\}} \leftarrow \gamma \cdot \text{EMA}_{m_i}^{\{\mu\}} + (1 - \gamma)\mu(I_i')$
15:         $\text{EMA}_{m_i}^{\{\sigma\}} \leftarrow \gamma \cdot \text{EMA}_{m_i}^{\{\sigma\}} + (1 - \gamma)\sigma(I_i')$
16:     **EMA loss:** $\mathcal{L}_E = WD(\text{EMA}_{m_i}^{\{\mathcal{H}\}}, \mathcal{H}(I_i')) + \|\text{EMA}_{m_i}^{\{\mu\}} - \mu(I_i')\|_2^2 + \|\text{EMA}_{m_i}^{\{\sigma\}} - \sigma(I_i')\|_2^2$
17:     Compute normalized gradient map $G(I_i')$
18:     **Gradient loss:** $\mathcal{L}_G = \|G_i - G(I_i')\|_2^2$
19:     **Total loss:** $\mathcal{L}_1 = \mathcal{L}_N + \mathcal{L}_G + \mathcal{L}_E$
20:     Backpropagate $\mathcal{L}_1$ and update $\Phi$ parameters
21: **end for**
22: **return** trained CDM model $\Phi$

---

---

**Algorithm 2** Target-Specific Fine Harmonization

---

**Input**: Source MRIs $\{I_X\}$, Target MRIs $\{I_T\}$ (unpaired, containing same set of sequences)
**Parameter**: Pre-trained CDM $\Phi$ from Stage 1, BiomedCLIP encoder $\Psi$, DDIM steps $T_f$, $T_r$
**Output**: Fine-tuned CDM $\Phi^*$ for target-specific harmonization

1: **# Pre-processing (performed once):**
2:    **for** each MRI $I$ in $\{I_X\} \cup \{I_T\}$ **do**
3:       **Global Harmonization:** Use pre-trained $\Phi$ to harmonize $I$ to $I'$ via DDIM inference (FDP+RDP)
4:    **end for**
5:    **for** each globally harmonized image $I'$ **do**
6:       $I^0 = I'$
7:       **for** $t = 0 : T_f - 1$ **do**
8:          $I^{t+1} = \sqrt{\bar{\alpha}_{t+1}} I' + \sqrt{1 - \bar{\alpha}_{t+1}} \, \epsilon_\theta(I^t, t, G, m)$
9:       **end for**
10:      Save $I^{T_f}$ as the noisy image for fine-tuning
11:    **end for**
12: **#Target-sepcific Fine-tuning:**
13: **for** epoch $= 1$ to $N_2$ **do**
14:    **for** each source noisy image $I_X^{T_f}$, Globally harmonized source $I'_X$, Globally harmonized target $I'_T$, and original target $I_T$ **do**
15:       **# DDIM Reverse Diffusion:**
16:       $I_{X \to T}^{T_r} = I_X^{T_f}$
17:       **for** $t = T_r : 1$ **do**
$$I_{X \to T}^{t-1} = \sqrt{\bar{\alpha}_{t-1}} I'_X$$
18: 
$$+ \sqrt{1 - \bar{\alpha}_{t-1}} \, \epsilon_\theta(I_{X \to T}^t, t, G_X, m_X)$$
19:       **end for**
20:      Obtain translated MRI $I_{X \to T} = I_{X \to T}^0$
21:      **# CLIP-Style Embedding:**
22:      $S_T = \Psi(I_T) - \Psi(I'_T)$    {Target style embedding}
23:      $S_{X \to T} = \Psi(I_{X \to T}) - \Psi(I'_X)$    {Translated style embedding}
24:      **# Disentangled CLIP-Style Loss:**
25:      $\mathcal{L}_S = \|S_T - S_{X \to T}\|_1 + \left(1 - \frac{S_T \cdot S_{X \to T}}{\|S_T\|\|S_{X \to T}\|}\right)$
26:      **# Style Reconstruction Loss:**
27:      $I_T^{T_f}$ from $I'_T$ via DDIM FDP, $I_{T \to T}^0$ via DDIM RDP as above
28:      $S_{T \to T} = \Psi(I_{T \to T}) - \Psi(I'_T)$
29:      $\mathcal{L}_R = \|S_T - S_{T \to T}\|_1$
30:      **Total Loss:** $\mathcal{L}_2 = \mathcal{L}_S + \mathcal{L}_R$
31:      Backpropagate and update $\Phi$ parameters using $\mathcal{L}_2$
32:    **end for**
33: **end for**
34: **return** fine-tuned model $\Phi^*$

---

# 2 ADDITIONAL RESULTS

## 2.0.1 ADDITIONAL VISUALIZATION ON SRPBS

We present additional results on T1 MRIs from the SRPBS test set, showing visualizations of two subjects across 11 sites and difference maps between each method's harmonized MRIs and the ground truth target (site COI). The Baseline denotes the unharmonized raw MRIs.

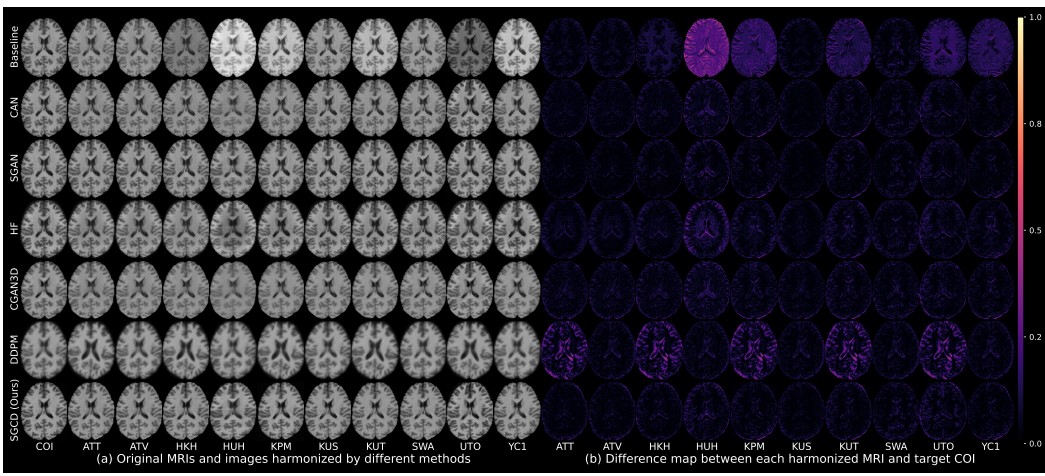

Figure 1: Axial view sample visualization results (a) and difference maps (b) of Subject 8 across 11 sites in SRPBS.

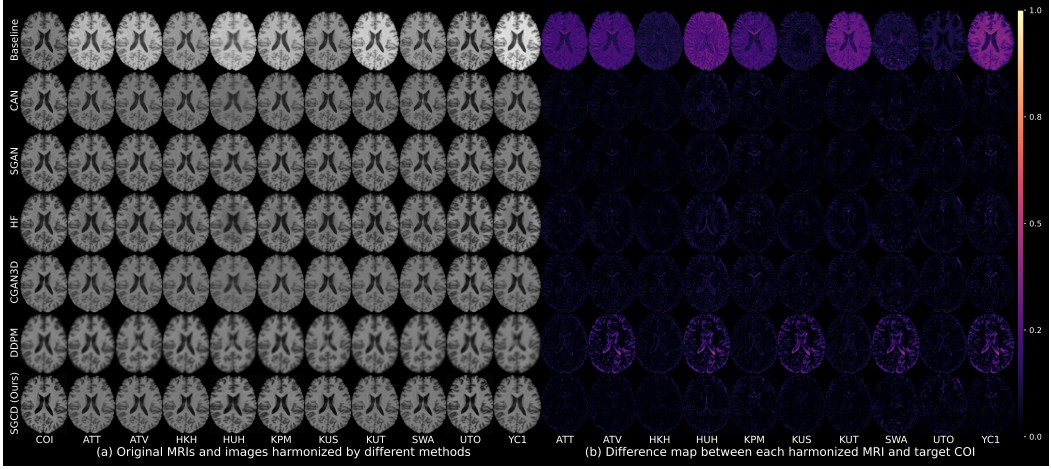

Figure 2: Axial view sample visualization results (a) and difference maps (b) of Subject 9 across 11 sites in SRPBS.

### 2.0.2 ADDITIONAL VISUALIZATION ON DWI-THP

We present additional results on T2 MRIs from the DWI-THP test set, showing visualizations of two subjects across 8 sites and difference maps between each method's harmonized MRIs and the ground truth target (site UCI). The Baseline denotes the unharmonized raw MRIs.

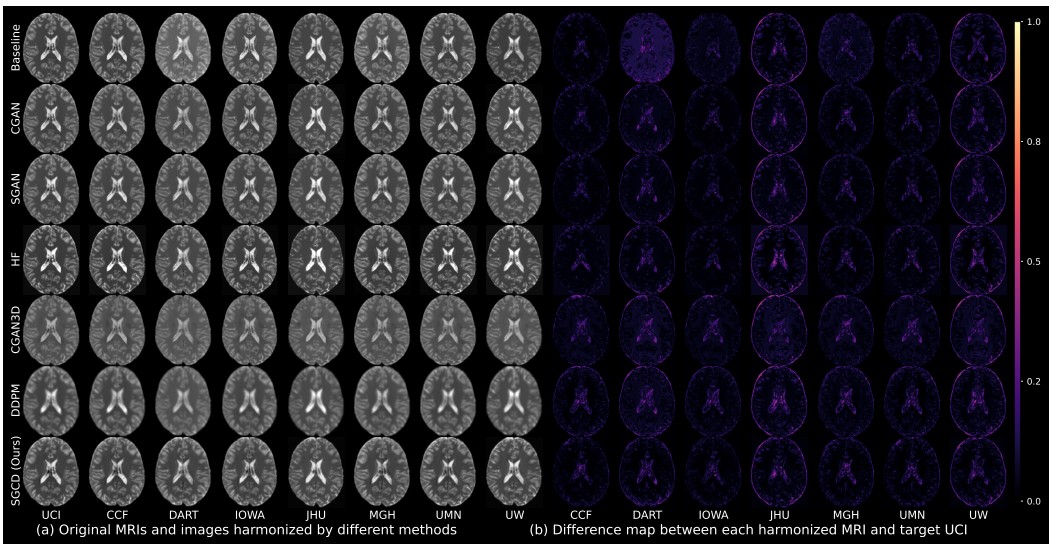

Figure 3: Axial view sample visualization results (a) and difference maps (b) of Subject THP0004 across 8 sites in DWI-THP.

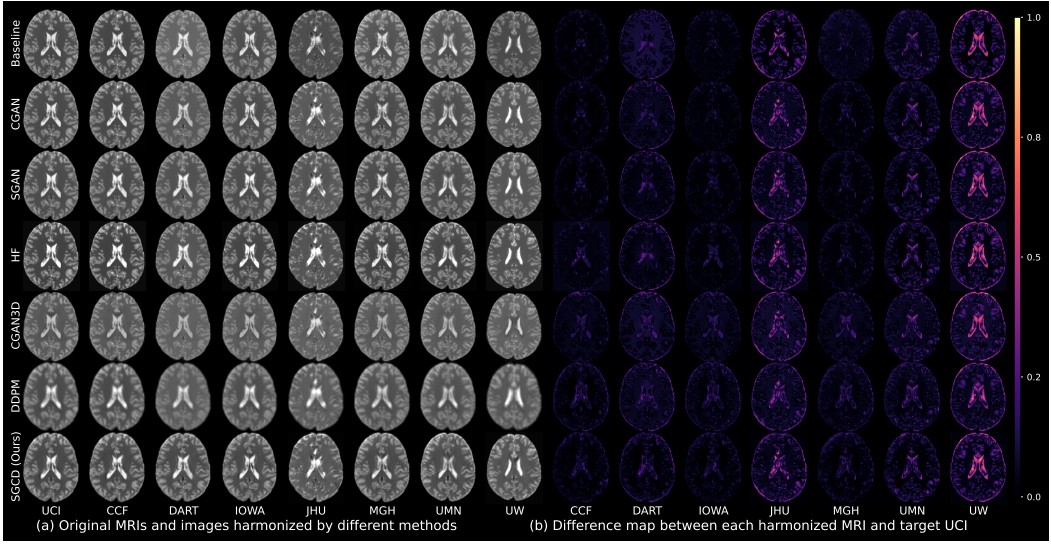

Figure 4: Axial view sample visualization results (a) and difference maps (b) of Subject THP0005 across 8 sites in DWI-THP.

