# OpenReview forum: "Harmonizing Multi-Site Multi-Sequence Brain MRI via Semantic-Guided Conditional Diffusion"
_ICLR.cc/2026/Conference — Submitted to ICLR 2026_

### Official Review · Reviewer_bDEP · 2025-10-23

**Soundness:** 3
**Presentation:** 3
**Contribution:** 3
**Rating:** 6
**Confidence:** 4

**Summary:**

This paper presents a diffusion-based method to remove non-biological differences in MRI images due to differences in scanners, and to adapt the images to match the style for a particular scanner site. The method does not require paired data (one individual scanned in multiple scanners), which is difficult to obtain. The paper compares to other methods and shows that they (a) better preserve anatomical detail according to tests on paired data,  while also (b) better removing site variation as measured by voxel intensity histograms and downstream site classifiers. The method incorporates several interesting features including spatial gradient map conditioning in the diffusion model, voxel histogram alignment loss, and a semantic loss incorporating BioMedCLIP embeddings to better preserve anatomical details while harmonizing images. An ablation study confirms the utility of each component.

While the paper is focused on neuroimaging and therefore may not appeal to the wider ICLR community, the authors describe the general problem of disentangling spurious and meaningful differences in brain images well and give some interesting and generalizable methods to solve the problem.

**Strengths:**

- Multi-site harmonization is an important challenge in neuroimaging. The authors describe the problem well. Diffusion based methods are increasingly being used here to improve image quality.

- While I've seen several works using the DDIM inversion trick for neuroimaging, there were some interesting methodological innovations here, including the use of spatial gradient maps for anatomical conditioning, histogram matching loss, and fine-tuning with a semantic loss.

- Comparisons include reasonable 2d and 3d baselines, as far as I'm aware.

- The paired subject evaluation, voxel histograms, and downstream classification checks were a nice way to verify that anatomical features are preserved while scanner differences are removed. The results look quite strong.

**Weaknesses:**

- DDIM inversion presentation. I think there are some issues. Either I've misunderstood something or there are mistakes in the equations. My main issue is that the current notation suggests T (the target domain) is never used in the DDIM reversal. See specific questions below. (The notation T (target domain) vs. T_f, T_r (timestep indices) is also confusing.)


-  The ablation discussion was useful, but I didn't think the single result in Fig. 5a was enough to support all the comments. For instance, "SGDCw/oT... fails to capture local style features such as tissue contrast and texture", how are we supposed to see something like this?

**Questions:**

1. I believe that the CLIP style guidance is only used for fine-tuning the model to preserve semantic detail during alignment. If so, is the backprop done through multiple layers of DDIM inversion, or just one (I hat)?
2. In Equations 10–11, should the sign before the predicted noise term be negative (as in standard DDIM) to represent noise removal?
3. Is the target domain T ever explicitly used in the sampling process, or is it only involved in the CLIP-based style loss?
4. I believe Eq. 10 and 11, and Alg. 2, line 16 and 17 are not correct. I would have thought that we get the "noise" or "latent space representation" using T, then we generate using the noise T but the gradient maps from X. Is this correct? If not please clarify as I have misunderstood how the domain target is meant to be incorporated. (And if I'm correct, please check and fix the notation.)
5.	Could you add references or discussion of prior uses of gradient consistency or gradient-difference losses for preserving anatomical detail? This is an interesting feature, but I hadn't come across the idea that it removes spurious site specific artifacts (though that seems reasonable, if we believe those are just random biases at each voxel.)
6. Out of curiosity, is it correct that the method cannot perform unconditional generation (since gradient conditioning requires an existing input anatomy)?
7. Is the L_R loss really necessary? It seems like it just measures the error in the DDIM ODE discretization. With more steps I'd expect this to go to zero. Still, I guess it can't hurt, and maybe it's useful to have fewer steps. Actually, you talk about tuning these parameters, but shouldn't results always improve with more steps?
8. Other work
One paper that may be relevant (though I didn't study it, so perhaps not)
https://arxiv.org/pdf/2409.00807v1.  Also, I feel that I've seen the "DDIM inversion" trick used on brain images in other contexts, like this one (https://ieeexplore.ieee.org/abstract/document/10782737/), for example. If there are one or two relevant examples they may be nice to mention or to contrast with your approach.

---

> ### Author Response · Authors · 2025-11-14
> **Response to Reviewer bDEP (Part 4/5)**
>
> 1	Question about model update strategy during second-stage finetuning
>
> We thank the reviewer for this insightful question. The reviewer is correct; the CLIP style guidance is indeed only used during the **Target-Specific Fine Harmonization (Stage 2)**.
>
> The backpropagation path is structured as follows: (1) **Forward Diffusion ($T_f=35$ steps)**: The CDM is used to gradually add learned noise to the globally harmonized input. This process is done **without loss calculation or gradient tracking.** (2) **Reverse Diffusion ($T_r$=25 steps):** Ideally, backpropagation would track gradients through all $T_r$ steps for the final loss calculation. However, accumulating the loss graph for $T_r=25$ reverse steps in a pixel-level model required **unmanageable GPU memory**, which is a constraint resulting from our architectural shift. (3) **Truncated Fine-Tuning Solution:** Our experiments showed that loss updates on early, very noisy reverse steps (e.g., t=25 down to t=15) led to unstable fine-tuning. Therefore, we utilize a **truncated diffusion fine-tuning** strategy. We perform the early reverse diffusion steps without gradient tracking, and then, **only on the last T=10 steps**, we calculate the loss and perform backpropagation at every step. This balances memory efficiency with stable convergence.
> This implementation detail was omitted from the initial submission due to space constraints. We plan to add this detail to the implementation section of our revised manuscript.
>
> 2	Question regarding incorrect Equation 10 and Equation 11
>
> We appreciate the reviewer’s query regarding our specialized FDP and RDP equation during the second-stage finetuning.
> Equation 10 is a **reformulation of the standard FDP defined in our Stage 1 (Equation 1)**, adapted specifically for the style fine-tuning objective. The purpose of Stage 2 FDP is to deterministically generate an informative noisy latent code ($I_{X}^{T_f}$) suitable for the RDP to start the style transfer. We achieve this by making two logical substitutions to the standard FDP process: (1) Instead of starting from the raw MRI ($I_i^0$), we start from the **Globally Harmonized MRI ($I_{X}^{\prime}$)**, as this is the "clean" state for the fine-tuning stage. (2) Instead of adding generic sampled noise ($\epsilon$), we add the **learned, CDM-predicted noise ($\epsilon_{\theta}$)** based on the current noisy state ($\mathbf{I}_{X}^{t}$). This formulation is a specialized deterministic FDP used to iteratively generate the target noisy state for style fine-tuning. We will clarify the text to ensure this derivation from Equation 1 is explicit in the final manuscript.
>
> Equation 11 does have a variable notation error, which may have caused confusion. The current $I’_X$ term is incorrect and should be replaced by $\hat{I}_X$, which is the dynamic, one-step predicted clean image estimate at each step t (see Equation 4). With this term corrected, our Equation 11 is a direct re-formulation of Equation 12 in the original DDIM paper [2] (with $\sigma_t=0$). The plus sign is therefore correct in this standard RDP formulation.
>
> 3	Question regarding target-domain usage.
>
> The MRIs in the target domain T are **only involved in the CLIP-based style loss**, which effectively guides the iterative sampling process without requiring T to be a direct input condition to the CDM.
>
> 4	Question regarding Equation 10&11 error and the usage of the target domain
>
> As we answered in the previous question 4.2, Equation 10 (FDP) is a correct specialized deterministic FDP used to iteratively reintroduce noise for the style fine-tuning process. While Equation 11 is incorrect in the term notation. The lines in Algorithm 2 will be corrected to reflect the use of the dynamic term $\mathbf{\hat{I}}_{X}$ in the reverse step to ensure consistency with the corrected equations.
>
> **Clarification on target domain incorporation:** The reviewer may have misunderstood the implicit role of the target domain T in our framework. The target domain T is **not used as a conditioning variable** in the DDIM sampling process (Equations 10/11). The only explicit conditions are the gradient maps from X ($G_X$) and the sequence label ($m_X$). The target domain T is **only involved in the CLIP-based style loss ($L_{S}$)** in Stage 2. The target image $I_T$ and its globally harmonized counterpart $I_{T}^{\prime}$ are used to extract the **Target Style Embedding ($S_T$)**. This $S_T$ then acts as a loss constraint, **implicitly guiding the backpropagation through the DDIM steps** to ensure the translated image matches the target style T. We do not mix noise/latent codes from T with gradients from X. This would violate the disentanglement goal, potentially transferring unwanted structural details from T to X. Our method strictly maintains the anatomical content from X (via $G_X$) while only transferring the style information dictated by the $S_T$ loss.

---

> ### Author Response · Authors · 2025-11-14
> **Response to Reviewer bDEP (Part 5/5)**
>
> 5	Question regarding the percentile-normalized gradient map condition
>
> The idea of using gradient information (Gradient Difference Loss, GDL) to measure the anatomical consistency is well-established in many medical image synthesis work [3], which is also utilized in our previous MICCAI work [1] as a secondary loss constraint to ensure the content preservation.
>
> However, our current ICLR submission utilizes a fundamentally redesigned gradient approach: it uses the gradient map as a **primary, style-agnostic input feature** to the CDM. The reviewer is correct that a **raw gradient map** is not inherently style-free, as a sharper image will have overall larger gradient absolute values. Our claim of "style-free" refers to the **percentile-normalized gradient map** used as the CDM input. The process achieves this by: (1) We first calculate the **99th percentile** of the absolute gradient value within each image. This percentile value acts as a robust, image-specific proxy for the upper bound of the "strongest" edge magnitude, while ignoring potentially large outlier pixels. (2) We then divide the entire gradient map by this 99th percentile value. This step removes the image-specific global scaling factor, transforming the map into a representation of the **relative strength of edges and textures**. The resulting map preserves the structure while effectively discarding the scanner-induced magnitude information (the **contrast profile**), making it style-agnostic.
>
> 6	Question on unconditional generation
>
> Yes, the reviewer is correct. The SGCD framework proposed in this work is fundamentally designed as a conditional generative model for unpaired MRI harmonization (unpaired image-to-image style translation). The input structural conditioning is essential for preserving anatomical details during the harmonization process.
>
> 7	Question on style reconstruction loss ($\mathcal{L}_R$) and finetuning diffusion steps
>
> We thank the reviewer for their critical questions regarding $L_{R}$ and the parameter tuning.
> The $L_{R}$ is necessary as it measures the difference between the Target Style Embedding ($S_T$) and the Target-to-Target Translated Style ($S_{T \to T}$). It essentially checks the cyclical consistency of style within the target domain. This loss is necessary to validate that the model has truly learned to capture the style distribution of the target domain $T$. Ideally, when a target MRI is passed through a well-trained harmonizer, it should return the target scan without significantly distorting its content or style, which is crucial for achieving stable harmonization across sites.
>
> The reviewer is correct that theoretically, results should improve with more steps for ODE approximation. However, in practical training involving deep networks and resource constraints, this is not always the case. As increasing the FDP and RDP steps ($T_f$ and $T_r$) significantly increases the training and inference time, making the model validation and tuning impractical. Our ablation study (Fig. 5(b)) shows that model performance (SSIM/WD) is not purely monotonic with step count. We found that performance peaks where $T_f$ and $T_r$ are balanced, specifically when $T_f=35$ and $T_r=25$. This is because excessive steps can lead to **over-sampling of the noise model** or introduce slight artifacts (over-harmonization). Therefore, tuning these parameters is necessary to find the optimal trade-off between image fidelity and practical computational efficiency.
>
> 8	Question on related work suggestions
>
> We thank the reviewer for pointing out the highly relevant literature. We will study the suggested works on DDIM inversion and incorporate them into the Related Work section of our revised manuscript.

---

> > ### Comment · Reviewer_bDEP · 2025-11-25
> > **Acknowledgement**
> >
> > Thanks for the detailed responses, and the careful comparison with the MICCAI submission.

---

### Official Review · Reviewer_qmeS · 2025-10-26

**Soundness:** 2
**Presentation:** 3
**Contribution:** 1
**Rating:** 2
**Confidence:** 4

**Summary:**

The presented work researches the use of diffusion models to harmonize MR images acquired with different MR scanners. In the first stage, the proposed method translates images to a sequence-specific, site-invariant representation. This is achieved by processing the images via a diffusion model, which is conditioned on edge maps from the original image as well as a learnable soft intensity histogram. In the second stage, the obtained unified representations are translated to the target style using yet another conditional diffusion model. The conditioning for the site-specific image appearance is generated by obtaining embeddings from the target image and its unified representation via a pre-trained CLIP encoder and calculating their difference. In experiments using three MR image datasets, the proposed method is shown to successfully translate images to diverse target styles, which is quantified via image similarity metrics, comparison of tissue segmentation maps, and measuring the utility of harmonized MR images to train age prediction models.

**Strengths:**

- MR scanner harmonization is a topic of emerging scientific interest, with only a limited number of related works researching the use of deep learning for this purpose. Moreover, any successful method would have clear practical benefit for scientists and clinicians handling MR images acquired with different scanners.

- The proposed method is intuitively understandable, with the use of a pre-trained CLIP encoder to disentangle content and style as well as a learnable soft-histogram to guide the site-invariant representation learning being interesting and potentially novel.

- The manuscript is clearly structured, illustrated and well written, making it easy to understand and follow.

- The proposed method is extensively evaluated by comparing it to five relevant baselines using three datasets. Moreover, the generated images are assessed through the lens of image similarity metrics, anatomical realism, and their utility as training data for neural networks for brain age prediction.

**Weaknesses:**

- There is considerable overlap with previously published work, which is most likely by the same authors (Wu, Mengqi, et al. "Unpaired Multi-site Brain MRI Harmonization with Image Style-Guided Latent Diffusion." International Conference on Medical Image Computing and Computer-Assisted Intervention. Cham: Springer Nature Switzerland, 2025). In particular, the entire second stage – called Target-Specific Fine Harmonization in the submitted work and CLIP-Style Fine Harmonizer in the published work – are essentially the same. Moreover, the use of gradient maps to guide the harmonization in the first stage is copied from the first work. As a result, the methodological novelty of the presented work is substantially diminished.

- While the ICLR’s call for papers explicitly mentions work exploring applications in healthcare, I feel that the presented study only makes limited contributions towards the general advancement of representation learning. I believe that the presented work with its highly specialized topic of MR scanner harmonization and tailored solution is mostly relevant for scientists working in the domains of medical image analysis or magnetic resonance imaging.

- The proposed method barely outperforms several baselines with regard to image similarly metrics, tissue structure, and downstream utility. These concerns are further amplified by ambiguities related to the tuning of the method’s hyperparameters (see further questions below).

- Circling back to the aforementioned similarities to previously published work, it is noteworthy, that the paper by Wu et al. reports markedly better performance of both the proposed method and baselines in apparently similar evaluation settings.

**Questions:**

In addition to above’s major concerns I have a couple of minor questions:

- The authors call the gradient maps “style-free”. However, I am unsure if these are truly style-free as a range of scanner-specific image characteristics, such as contrast, sharpness and grid homogeneity, will directly impact these gradient maps. I was wondering whether the authors have any quantitative results that back their claim that the gradient maps are unaffected by site effects?

- The authors introduce the variable $m_i$, which encodes sequence type as a set of discrete numbers. I was wondering whether this is truly being used in practice instead of encoding the sequence type as a one-hot encoding?

- It is unclear how several of the method’s hyperparameters were tuned. While the authors have compared different numbers of forward and reverse diffusion steps in an ablation study, they should also describe how the weighting of the many loss terms was determined.

- Additionally, the statement “we ensured consistent training hyperparameters across all methods for fair comparisons” is ambiguous. Does this mean that the same hyperparameters were used across all methods (leading to a potential disadvantage for baselines that may benefit from additional hyperparameter tuning) or that the same hyperparameters as in the original publication were used (leading to a potential benefit for baseline that may have originally been tuned for different data domains) or something entirely different?

**Details Of Ethics Concerns:**

As outlined in my comments to the authors, there is considerable overlap with previous work (Wu, Mengqi, et al. "Unpaired Multi-site Brain MRI Harmonization with Image Style-Guided Latent Diffusion." International Conference on Medical Image Computing and Computer-Assisted Intervention. Cham: Springer Nature Switzerland, 2025). Considering the highly similar method, experimental setup, and manuscript style, I suspect that both studies were conducted by the same group of authors.

In particular, the entire second stage – called Target-Specific Fine Harmonization in the submitted work and CLIP-Style Fine Harmonizer in the published work – is essentially the same. Moreover, the use of gradient maps to guide the harmonization in the first stage is copied from the first work. As a result, the methodological novelty of the presented work is substantially diminished. Crucially the previous work is not referenced or mentioned at all in the submitted paper.

Conversely, there are a few novel contributions in the submitted work, specifically the introduction of a soft intensity histogram to guide extraction of site-invariant MR image representations.

---

> ### Author Response · Authors · 2025-11-14
> **Response to Reviewer qmeS (Part 3/5)**
>
> 1. Question on style-free gradient conditions
>
> We thank the reviewer for this insightful comment. The reviewer is correct that a **raw gradient map** is not inherently style-free, as its magnitude is directly affected by scanner-specific characteristics like local contrast and sharpness; a sharper image will have overall larger gradient absolute values.
>
> However, our claim of **"style-free"** refers to the percentile-normalized gradient map used as the Conditional Diffusion Model (CDM) input. Our contribution is the percentile-based normalization strategy, which is designed to decouple structural information from scanner-specific intensity scaling and contrast profile. The process achieves this by: (1) We first calculate the **99th  percentile** of the absolute gradient value within each image. This percentile value acts as a robust, image-specific proxy for the upper bound of the "strongest" edge magnitude, while ignoring potentially large outlier pixels. (2) We then divide the entire gradient map by this 99th percentile value. This step removes the image-specific global scaling factor, transforming the map into a representation of the **relative strength of edges and textures**. The resulting map preserves the structure while effectively discarding the scanner-induced magnitude information (the **contrast profile**), making it style-agnostic.
>
> To validate this claim quantitatively, we can measure the distributional distance (e.g., Wasserstein distance, WD) between the normalized gradient maps of different sites and expect it to be significantly lower than the WD between the raw gradient maps. We commit to conducting this quantitative experiment and including the results in the final manuscript to formally validate the style-agnostic nature of our conditional input.
>
> 2	Question about sequence label condition
>
> We can confirm that the sequence type is used as a class condition in the Conditional Diffusion Model (CDM). In practice, the sequence label $m_i$ (e.g., 0 for T1w, 1 for T2w) is **not** used as a raw discrete input. Instead, we use an **embedding layer** to map the discrete numerical sequence label to a 128-dimensional, dense embedding vector. This embedding vector is then injected into the CDM's 3D U-Net via the **cross-attention mechanism** (alongside the time-step encoding) to allow the model to learn sequence-specific diffusion and denoising operations. We will include this detail in the implementation section of our revised manuscript.
>
> 3	Questions about loss weights tuning
>
> We employed a systematic, data-driven approach to select the final loss weights: Initially, all loss weights were adjusted to ensure each loss term contributed equally to the total gradient magnitude, establishing a stable starting point for training. The final weights were then determined through leave-one-subject-out cross-validation on the DWI-THP training dataset. We evaluated the configuration by calculating the mean inter-image (cross-site) Wasserstein Distance (WD) loss on the validation folds. We systematically adjusted weights, prioritizing the configuration that yielded the best (lowest) validation WD score across both T1w and T2w modalities, ensuring the final configuration was empirically validated to prioritize effective style alignment.
>
> 4	Question about baseline methods training
> Our statement "we ensured consistent training hyperparameters across all methods for fair comparisons" refers to a balanced approach to architectural and training parameters: (1) We honored the **default structural hyperparameters** (e.g., number of layers, block types, normalization methods, initial learning rate) provided by each competing method's original code. This ensures that we evaluate the **core design and architecture** of each method fairly. (2) As for the training hyperparameters, since methods use different implementations (step-based vs. epoch-based), we ensured a comparable **total number of global steps** across all methods and trained until convergence. We used validation curves to select the optimal checkpoint for each baseline, honoring early stopping if implemented, or manually selecting the best checkpoint based on validation performance. These represent our sincere effort to uphold fairness.

---

> > ### Comment · Reviewer_qmeS · 2025-11-27
> > **Response to the authors' rebuttal**
> >
> > Regardless of any textual differences between the presented work and paper previously published at MICCAI [1], I remain unconvinced regarding the technical novelty of the presented study. In my opinion the change from latent to pixel-level diffusion, inclusion of multiple input sequences at once, and slight expansion of experimental scope only constitute very minor changes. Crucially, none of these changes are used to motivate the ICLR study, even if the MICCAI work was properly. Instead, the ICLR study is primarily motivated by the introduction of the two-stage harmonization approach that extracts a site-independent latent representation that can be subsequently adapted to different site-specific contrasts, just as the previous MICCAI paper.
> >
> > I leave the further evaluation whether this degree of overlap between the works without proper attribution constitutes plagiarism to the area and program chairs. However, I firmly stand by my previous recommendation to reject the paper.
> >
> > [1] Wu, Mengqi, et al. "Unpaired Multi-site Brain MRI Harmonization with Image Style-Guided Latent Diffusion." International Conference on Medical Image Computing and Computer-Assisted Intervention. Cham: Springer Nature Switzerland, 2025

---

### Official Review · Reviewer_copw · 2025-11-10

**Soundness:** 2
**Presentation:** 2
**Contribution:** 1
**Rating:** 0
**Confidence:** 5

**Summary:**

The content of this work appears to echo that of a paper titled, "Unpaired Multi-Site Brain MRI Harmonization with Image Style-Guided Latent Diffusion" accepted to MICCAI 2025. The paper cites another work by the author of the MICCAI paper, but not the MICCAI paper itself, though the figures, datasets, writing, baselines, and evaluation methods are almost identical. I believe that the only novel contribution from this work is section 3.1.2, the EMA-based record update in this particular context.

**Strengths:**

N/a.

**Weaknesses:**

The figures seem to be slight modifications of the MICCAI paper that bear obvious resemblance, including the same order. The numerical values in the tables are shown in a different format with similar values and the same metrics in the same order (age and site classification scores for all baselines are identical except for rounding variation to the other paper except for the proposed method). Both studies used exactly 3,984 subjects across 58 sites and the same competing methods. I can provide additional examples if necessary.

**Questions:**

I do not have any questions regarding this work at this time.

**Details Of Ethics Concerns:**

Substantial mirroring of a paper available online from MICCAI 2025 (see paper weaknesses).

---

### Comment · Area_Chair_67s2 · 2025-11-12
**Near Duplicate Work Found**

Submission 7827 Authors:

All three reviewers have identified a duplicate or near-duplicate work published at another conference. I am therefore taking this step immediately, instead of waiting for discussion:

1. Please explain the differences between this submission and the MICCAI paper, and why the MICCAI paper was not cited.
2. Please enumerate exactly why this submission is not a double submission/self-plagiarism, and its relative novelty.

The SAC is also aware of this situation.

---

> ### Author Response · Authors · 2025-11-14
> **Defense Against Dual Submission/Self-plagiarism (Part 1/5)**
>
> Dear Area Chair and Senior Area Chair,
>
> We sincerely appreciate all **three** reviewers for their constructive comments. We thank the Area Chair for providing the opportunity to clarify the concerns raised by two of the three **Reviewers (copw and qmeS)** regarding potential dual submission/self-plagiarism with the published MICCAI 2025 paper, "Unpaired Multi-Site Brain MRI Harmonization with Image Style-Guided Latent Diffusion" [1]. We respectfully address this serious concern and hereby offer the following detailed explanation and **technical defense**, asserting that our submission is **not a dual submission or self-plagiarism**.
>
> **1	Explanation for the Missing Citation and Differences**
> - **Timing:** The MICCAI paper was officially published online on September 21st, 2025, just three days before the ICLR 2026 submission deadline on September 24th, 2025. In the immediate rush to finalize the current ICLR manuscript and meet the deadline, we failed to update the reference list to reflect the paper's final publication status. We assure the Area Chair and reviewers that we treat publication ethics with the utmost gravity, and we commit to fully rectifying this oversight.
> - **Textual integrity:** The overall document similarity between this ICLR submission and the MICCAI paper, measured by iThenticate (from the Crossref source), is 10%, confirming the extensive rewriting of this manuscript.
>
> **2	Technical defense against double submission/self-plagiarism**
>
> This ICLR submission is not a self-plagiarism or duplicate submission, but rather focuses on a fundamental shift in model architecture, new methodological components, and new experimental validation. Details are introduced below.
>
> 2.1 Architectural Shift: Latent Diffusion vs. Pixel-level Diffusion
>
> The core methodological difference is a shift in the diffusion model architecture and overall training approach. The previous work [1] utilized a **Conditional Latent Diffusion Model (CLDM)**, which requires an initial 3D autoencoder to encode/decode MRIs through a low-dimensional latent space. While latent space harmonization generally reduces the computational cost, we observed that this came at the expense of introducing image blurriness due to autoencoder compression. Consequently, in the current submission, we completely redesigned the underlying architecture to a **Pixel-Level Conditional Diffusion Model (CDM)** that operates directly on the raw image pixel space to preserve image fidelity.
>
> 2.2 Novel Methodological Contributions (Multi-Sequence Handling)
>
> Beyond the architectural shift, the ICLR submission introduces a new capability for handling multi-sequence input (T1w and T2w) simultaneously during one training. This greatly expanded the generalizability of our MICCAI work, which only supports one input sequence at a time. This feature is not a simple addition to the model input but requires the development of three novel components:
>
> (a)	Sequence-specific conditioning:
>
> The model is now conditioned on the sequence label ($m_i$) to differentiate the T1w and T2w (can be potentially scaled to more sequences) input during the training. The numerical sequence label is first mapped into a sequence embedding through an embedding layer and later introduced to the CDM via a cross-attention mechanism to allow the model to learn sequence-specific adaptation.
>
> (b)	Style-free anatomical conditioning:
>
> The mechanism for providing anatomical content information represents a significant methodological redesign from our prior work. Our previous MICCAI paper utilized the input latent map after Instance Normalization (IN) as the content condition, which provided a coarse style removal in the compressed latent space. In the current SGCD submission, operating in the pixel space and to support multi-site, multi-sequence input, we required a more robust solution. We therefore fundamentally changed the feature input: we now use the percentile-normalized gradient map of the raw MRI directly as a primary input condition to the Conditional Diffusion Model (CDM). This percentile-normalization is key to removing the influence of small outlier voxels and absolute edge strength differences due to site-related intensity or contrast variations, while preserving the relative tissue boundary strength within each MRI. This role is a qualitative shift in purpose: **unlike the MICCAI work**, which utilized the gradient map merely as a loss constraint for content preservation, the SGCD framework leverages the percentile-normalized gradient map as a primary condition feature that actively guides the generative process, ensuring highly precise anatomical fidelity during the multi-sequence alignment.

---

> ### Author Response · Authors · 2025-11-14
> **Defense Against Dual Submission/Self-plagiarism (Part 2/5)**
>
> (c)	EMA-based record update
>
> Though both our MICCAI and ICLR works contain a two-stage training scheme because we believe it is effective for the unpaired harmonization task. However, the core mechanism and objective of **Stage 1 (Coarse/Global Harmonization)** are substantially different. In the MICCAI paper, the first-stage coarse style alignment relied entirely on **implicit alignment** within the latent space. By using Instance Normalization (IN) on latent maps as a style-suppressed condition and training the CLDM on diverse multi-site T1w samples, the model was expected to implicitly learn to generate unified latent codes.
>
> However, introducing **multi-sequence input** (T1w and T2w) in the ICLR work, which drastically increases the disparity in pixel intensity distributions, meaning that we could no longer rely on implicit alignment.  We addressed this by introducing an **Exponential Moving Average (EMA)-based style record** explicitly for each sequence type. This is the **core novelty of the new Stage 1**. These sequence-specific EMA records track the current moving average of intensity via fully differentiable **soft-histograms**, global voxel mean, and standard deviation. These records are updated on-the-fly and used to **explicitly guide the model training** through an EMA loss, ensuring robust and measurable alignment across distinct sequence domains.
>
> (d)	Refined target-specific fine-tuning
>
>  While the core concept of utilizing BiomedCLIP embeddings for semantic style guidance remains identical, Stage 2 (Target-Specific Fine Harmonization) required non-trivial re-implementation due to the latent-to-pixel level architectural shift. The current pixel-space CDM operates in a much larger dimension than the previous latent model, and we discovered that accumulating the gradient graph for the full DDIM reverse diffusion steps ($T_r=25$) during fine-tuning led to unmanageable GPU memory consumption. To solve this architectural constraint, we implemented a truncated diffusion fine-tuning loss update approach that performs the loss update only on the last T=10 steps (selected based on SNR analysis) of the reverse diffusion process. While this crucial implementation detail was omitted from the initial submission due to space constraints, it underscores the complexity introduced by the pixel-space architecture and demonstrates the necessary effort required to achieve stable training.
>
> 2.3 Non-trivial expansion of experimental scope
>
> We acknowledge the reuse of core datasets and competing baselines, which is necessary to position the SGCD method (ICLR) against the established SOTA methods. However, the experimental section required **non-trivial expansion and re-implementation** to validate the new multi-sequence capability. Unlike the MICCAI paper, which focused solely on T1w MRIs, the ICLR submission is designed for and validated on **multi-sequence data (T1w and T2w)** from the DWI-THP dataset. We include new T2w harmonization results in Figure 2 (b), Figure 3(b), and in the Appendix. Furthermore, we have included new ablation studies (Fig. 5(a)) to validate each component of the redesigned model architecture, including variants without the sequence-specific EMA-based constraint, without the gradient-based anatomical conditioning, with only global harmonization, and a variant with a generic CLIP encoder. The results of these new ablations (Fig. 5(a)) highlight the effectiveness of our newly introduced components, clearly suggesting that this is a non-trivial expansion of our previous studies.
>
> Therefore, this ICLR submission represents a **non-trivial methodological advance** in the field of brain MRI harmonization. Our work contains substantial technical novelty, including: the fundamental **architectural shift** from latent- to pixel-level diffusion; the re-designed **style-free anatomical conditioning** using a percentile-normalized gradient map; the novel **EMA-based multi-sequence component**; and the **refined, memory-efficient fine-tuning strategy** required by the new architecture. We also confirm that the manuscript text is **substantially different from the prior MICCIA paper** (with an overall iThenticate score of 10%), proving that this was not a textual reuse.
>
> We believe that this work does **not constitute a breach of research integrity**. In the Discussion phase, we will fully cite the MICCAI paper and expand the differentiation of the technical sections in the final manuscript.

---

### Meta-Review · Area_Chair_4iqx · 2025-12-25

**Summary:**

The submission proposes a "Semantic-Guided Conditional Diffusion" (SGCD) framework for multi-site, multi-sequence MRI harmonization. While the problem is clinically relevant, the review process uncovered that this work is a near-duplicate of a paper published by the same authors at MICCAI 2025, just days before the ICLR deadline. All reviewers and the (initial) Area Chair identified substantial overlap in methodology (diffusion-based harmonization, gradient guidance, CLIP loss) and experimental design. The consensus is that the submission violates dual submission policies and offers insufficient technical novelty (pixel vs. latent diffusion) to justify a separate publication at a top-tier venue.

**Reviewer Concerns:**

The authors provided clarifications regarding technical details, specifically the "style-free" nature of percentile-normalized gradient maps and the implementation of sequence embeddings.

The most critical outstanding concern is the extensive overlap with the authors' MICCAI 2025 paper, which was not cited in the original submission.

Reviewers (qmeS, copw, and later bDEP) agreed that the modifications (switching from Latent to Pixel diffusion, adding multi-sequence support) represent minor engineering tweaks rather than the "non-trivial expansion" claimed by the authors.

The improvements over baselines were viewed as marginal, and the "new" contribution was seen as getting "double credit" for the same core research.

**Reviewer Scores:**

Reviewers copw (Score: 0) and qmeS (Score: 2) have already strongly recommended rejection based on the overlap and lack of novelty. Reviewer bDEP (Score: 6) initially leaned towards acceptance but, after discovering the MICCAI paper during the discussion phase, explicitly noted the "substantially overlapping published work" and agreed that this constitutes "double credit". Therefore, if able to update, bDEP would likely lower their score to align with the unanimous rejection consensus.

---

### Decision · Program_Chairs · 2026-01-26

Reject